# Microbial Population Succession and Community Diversity and Its Correlation with Fermentation Quality in Soybean Meal Treated with *Enterococcus faecalis* during Fermentation and Aerobic Exposure

**DOI:** 10.3390/microorganisms10030530

**Published:** 2022-02-28

**Authors:** Hao Ma, Weiwei Wang, Zhenyu Wang, Zhongfang Tan, Guangyong Qin, Yanping Wang, Huili Pang

**Affiliations:** 1School of Physics and Microelectronics, Zhengzhou University, Zhengzhou 450052, China; mahaoworks@foxmail.com (H.M.); bingzhi213608@163.com (W.W.); 2Henan Key Lab Ion Beam Bioengineering, School of Agricultural Sciences, Zhengzhou University, Zhengzhou 450052, China; 18438699518@163.com (Z.W.); tzhongfang@zzu.edu.cn (Z.T.); qinguangyong@zzu.edu.cn (G.Q.); wyp@zzu.edu.cn (Y.W.)

**Keywords:** optimization, soybean meal, fermentation, microbial community

## Abstract

This study assessed the effects of *Enterococcus faecalis* (*E. faecalis*) in combination with protease on fermentation characteristics and microbial communities during ensiling and aerobic exposure phases of soybean meal (SBM). In this study, response surface methodology (RSM) was used to optimize the optimal growth conditions of *E. faecalis* ZZUPF95, which produced protease, and fermented SBM under the optimal fermentation conditions. The fermentation test was divided into four groups as follows: CK (Control check), ZZUPF95, Protease and ZZUPF95+Protease groups. Results showed that the best medium ratio of ZZUPF95 was glucose 1%, peptone 2%, inorganic salt 1.47%; fermentation time 36 h, inoculation amount 10%, ratio of material to water 1:1 is the optimal fermentation scheme; after fermentation and aerobic exposure treatment, ZZUPF95 and ZZUPF95 + Protease group can reduce the pH of feed, improve the content of lactic acid in the fermentation system, and have the effect of inhibiting the reproduction of pathogenic bacteria, increasing the content of crude protein and ether extract, and degrading crude fiber; the microbial community of SBM were changed after fermentation and aerobic exposure. This study explored the changes of fermentation quality of SBM, which has certain theoretical value to improve the fermentation mode and storage of SBM.

## 1. Introduction

In recent years, due to the changes in people's diets, the breeding industry has developed rapidly. Accordingly, feed materials, especially high-quality protein feed, have been studied widely. As an adequate plant protein raw material, soybean meal (SBM) has attracted extensive attention because of its high protein content, abundant amino acids essential for animals, and many functional nutrients such as fatty acids, vitamins, peptides, isoflavones, minerals, flavonoids, phenolic acids and saponins [1,2]. However, the macromolecular protein and antinutritional factors (ANFs) in SBM, such as trypsin inhibitor exogenous lectin and α amylase inhibitor, negatively affected the growth performance and intestinal health of herbivores, especially in young animals [3,4].

In order to improve the nutritional quality of SBM, enzymatic hydrolysis, heating, soaking and microbial solid-state fermentation are often used to degrade the size of SBM and ANFs [5]. However, heat stable ANFs are still unable to be degraded by heating, while excessive heating would destroy heat-sensitive amino acids, especially lysine and arginine denaturation, thereby reducing protein quality [6]. The study of Fasina et al. showed that most trypsin inhibitors and lectins were decomposed after heating at 90–110 °C for 20 min, but there are little effects on soybean globulin, β-concomitant soybean globulin and phytic acid with high thermal stability [7]. And the application of a chemical process is limited by chemical residue and high processing cost [8]. Fermenting processes constitute an effective strategy to deactivate ANFs and degrade macromolecular proteins which in turn improve the nutritional value of plant protein sources [9]. Microorganisms are able to metabolize high-molecular weight proteins into smaller, biologically active peptides or amino acids through the production of proteases [10]. Thus, fermented soybean meal (FSBM) showed a better feed efficiency than other methods.

As the most frequently used probiotics, Lactic acid bacteria (LAB) play an important role in improving feed quality due to their rich enzymatic systems such as peptidoglycan hydrolase, glycoside hydrolase, protease and lipase. LAB with high intestinal adhesion can also produce antibacterial proteins and block the adhesion of harmful microorganisms in the intestinal tract [11,12]. Studies by Borgesi et al. [13] and Mukerji et al. [14] confirmed that LAB fermentation has positive effects on immune regulation and maintenance of intestinal colony balance. In addition, they contribute to improved resistance to diseases and infections [15]. In conclusion, LAB has great research potential in SBM fermentation.

In this study, a protease-producing *Enterococcus* (*E.*) *faecalis* strain ZZUPF (Zhengzhou University Pig Feces) 95 with excellent growth performance was used for SBM fermentation. The optimal growth conditions and fermentation process of the strain and the changes of microorganisms and fermentation quality before and after SBM fermentation and during aerobic exposure were investigated, which has certain theoretical value for improving the fermentation mode and storage of SBM.

## 2. Materials and Methods

### 2.1. Materials and Microorganisms

SBM was purchased from China Grain Storage Oil Co., Ltd. (Tangshan, China). The DM (dry matter) concentration for SBM was 892.3 g/kg, the CP (crude protein), CF (crude fiber) and EE (ether extract) contents were 43.5, 6.8 and 1.6 (%DM) for SBM, respectively. The populations of LAB, aerobic bacteria, coliform bacteria and bacilli of SBM were 6.5, 8.1, 5.1 and 4.7 lg cfu/g FM (fresh matter), respectively. *E. faecalis* ZZUPF95 with excellent protease-producing ability and has good inhibition abilities on *Staphylococcus aureus*, *Bacillus subtilis*, *Escherichia coli*, *Listeria monocytogenes*, *Pseudomonas aeruginosa* and *Micrococcus luteus*, was isolated from the fresh feces of healthy weaned piglets was activated in MRS (Man Rogosa Sharp) broth overnight at 30 °C.

### 2.2. Optimization of Growth Conditions for ZZUPF95

Growth conditions of ZZUPF95 were optimized by the single-factor tests (SFT) and response surface methodology (RSM). The type and content of carbon, nitrogen and content of inorganic salt were optimized by SFT for growth and acid production capacity of ZZUPF95. According to RSM experimental results, a three-factor and three-level experiment was designed using Design-Expert software (Stat-Ease, Inc., Minneapolis, MN, USA), and the optimal medium conditions were obtained by analyzing the experimental results. In addition, on the basis of an inoculation amount of 3%, ZZUPF95 was incubated in a different culture medium with different culture temperature, culture time and pH of medium; the OD (optical density)_600_ of bacterial suspension after culture was determined to screen out the best culture conditions.

### 2.3. Optimization of Fermented Soybean Meal (FSBM) Conditions for ZZUPF95

Based on the SFT experiment results, the orthogonal test with three factors and three levels was designed to obtain the optimal fermentation conditions. Taking the protease activity produced in ZZUPF95 fermented SBM as the response value, the effect of different fermentation time (24, 36, 48, 60 h and 3 d), ZZUPF95 inoculum amount (5, 7.5, 10, 12.5 and 15%) and material-water ratio (1:0.8, 1:0.9, 1:1, 1:1.1 and 1:1.2) on protein protease activity were measured by the Flint-phenol method [16].

### 2.4. Preparation of FSBM

ZZUPF95 was cultured in the above optimized MRS medium for 12 h, and the experiment was divided into four groups: (1) CK (Control check), SBM + water; (2) 95, ZZUPF95 + CK; (3) P, 1% protease (shanghai Lian Shuo Biotechnology Co. Ltd., Shanghai, China) + CK; and (4) P + 95, ZZUPF95 + 1% protease + CK. The ratio of feed to water and amount of ZZUPF95 added were based on the above optimized results. After the mixture, 300 g of each was added to sealing bags and vacuumized, each treatment had three replicates and all bags were fermented at room temperature. Samples were opened and taken on 12, 24, 36, 48, 60 h, and 3, 7, 12, 18, and 30 days (d) of fermentation, and aerobic fermentation was exposed to 3 d and 7 d after fermentation of 3 d (3–3 d, 3–7 d) and 30 d (30–3 d, 30–7 d), respectively, for analysis of fermentation quality, chemical composition and microbiology.

### 2.5. Analysis of FSBM

#### 2.5.1. Evaluation of Sensory 

The sensory evaluation was carried out with reference to the method of Meinlschmidt et al. [17]. After the FSBM samples were unsealed, sensory profiling was performed using descriptive sensory analysis: the samples were weighed into dishes, and the sensory evaluation team was composed of five people to evaluate the colors, odor and hand feel characteristics of SBM and FSBM during fermentation for 3 and 30 d, and aerobic exposure of 7 d by eye observation, nose smell and hand touch.

#### 2.5.2. Fermentation Quality

The pH of filtrate of FSBM was measured by a glass electrode pH meter (Mettler Toledo Co., Ltd., Greifensee, Switzerland), while lactic acid (LA) and acetic acid (AA) were further measured by HPLC (Waters 2695 HPLC system Waters Technology Co., Ltd., Milford, CT, USA) using a Symmetry C18 column (4.6 × 250 mm, 5 μm). The mobile phase was 25.4% vitriol with a flow rate of 0.6 mL/min, and the column temperature was 55 °C. Absorbance of the elution was monitored at 214 nm.

#### 2.5.3. Chemical Analysis

To determine the DM content, the samples were dried in a forced-air oven at 65 °C for 48 h to constant weight. CP, EE and CF were determined using the AOAC (Association of Official Analytical Chemists) standard method [18]; automatic Kjeldahl nitrogen, ether extract and crude fiber analyzer were used for determination, respectively.

#### 2.5.4. Microbiological Analysis

The icrobiological analysis referred to the method of Pang et al. [19]. Three bags were randomly selected and opened, then 10 g of fermentation sample from each bag were diluted with 90 mL of distilled water, and the supernatant was diluted serially to 10-fold and inoculated in triplicate on different agar plates. Yeast and mold colonies inoculated on PDA (potato medium agar) and incubation at 30 °C for 48 h. LAB colonies on MRS agar at 37 °C for 48 h. Coliform bacteria, clostridium difficile and bacillus, aerobic bacteria were incubated on EMB (Eosin Methylene Blue), CLO (Clostridium enrichment medium) and NA (Nutrient Agar), respectively, at 37 °C for 48 h.

#### 2.5.5. Scanning Electron Microscopy (SEM) of SBM and FSBM

Changes of FSBM (during fermented for 3 and 30 d) in the physical properties and microstructures were examined by scanning electron microscopy (SEM, Thermo Scientific/Helios G4 CX, Prague, Czech Republic) at ×2000-fold magnification, respectively.

#### 2.5.6. SDS-PAGE ((Sodium Dodecyl Sulfate-Polyacrylamide Gel Electrophoresis) Profile

SDS-PAGE was carried out to determine the degradation of protein in FSBM, and stacking and separating gels were prepared using 4% and 12% of acrylamide concentration, respectively. Samples were heated at 100 °C for 10 min prior to electrophoresis and the gels were stained with Coomassie Brilliant Blue R250. 

### 2.6. Bacterial and Fungi Community Analyses

#### 2.6.1. DNA Extraction

Changes in microbiota, including bacterial and fungal communities and structures, were analyzed during fermentation using high-throughput sequencing technology. The microorganism suspension was obtained by 20 g fermented SBM samples mixed with 180 mL of 0.85% (wt%) sterile NaCl solution and storing them in an inspissator at 100 rpm for 2 h. The suspension was filtered through a 0.22 mm sterile membrane and then placed in a 2 mL sterile microcentrifuge tube to extract DNA by DNA Kit (shanghai Lian Shuo Biotechnology Co. Ltd., Shanghai, China). Eluting the resulting DNA to a final volume of 80 µL, and the amount and concentration of DNA were evaluated using 1% agarose gel electrophoresis and the ratio of OD_260_ to OD_280_ nm. The OD was measured on a microplate reader (Thermo Scientific, Wilmington, NC, USA) and the eligible DNA samples were stored at −20 °C for further research.

#### 2.6.2. PCR Amplification

After the DNA extraction, amplification of the V3-V4 region of bacterial 16S rRNA using the primers 319F (5′-ACTCCTACGGGAGGCAGCAG-3′) and 806R (5′-GGACTACHVGGGTWTCTAAT-3′), while the primers Fits7 (5′-GTGARTCATCGAATCTTTG-3′) and ITS4 (5′-TCCTCCGCTTATTGATATGC-3′) amplify the ITS2 region of the fungi DNA. The amplification increment was 25 μL, including genomic DNA extract 25 ng, PCR premix 12.5 μL, primers 2.5 μL and water 25 μL. The PCR conditions were as follows: 98 °C for 30 s, 35 cycles of 98 °C for 10 s, annealing at 50 °C for 30 s (for bacteria) and 56 °C (for fungi), followed by a 45 s extension at 72 °C, and an additional 5 min extension at 72 °C. Each DNA sample was tested three times, and the PCR products obtained were combined. PCR products were separated by 1% agarose gel electrophoresis and purified by nucleic acid purification kit (Shanghai Lian Shuo Biotechnology Co. Ltd., Shanghai, China). The purified PCR products were subjected to high-throughput sequencing by Shanghai Applied Protein Technology Co., Ltd. 2.6.3 Bioinformatic analysis.

Sequence processing and analysis using QIME software (version1.9.0 http://qiime.org/scripts/assign_taxonomy.html accessed on: 21 September 2021). Sequences were assigned at >97% similarity using UPARSE. The qualified sequences were clustered into operational taxonomic units (OTUs) using the UPARSE pipeline with 97% similarity (version7.1 http://drive5.com/uparse/ accessed on: 25 September 2021), and using Qiime to determine the taxonomic characteristics of OTUs based on the database Silva (http://www.arbsilva.de accessed on: 26 September 2021) and Unite (ITS, http://unite.ut.ee/index.php accessed on: 26 September 2021). The sequencing data was visualized by cluster analysis based on the Jaccard distance matrix, and principal coordinate analysis (PCoA) based on the weighted UniFrac distance matrix, PCoA and heatmap analysis in R3.1.0 software.

### 2.7. Statistical Analysis

All experiments were performed three times. The results were processed for analysis of variance by SPSS software (version 21). The significant difference was determined at the *p* < 0.05 using Duncan’s multiple range tests.

## 3. Results

### 3.1. The Optimization of Inorganic Salt, Carbon and Nitrogen Source on Growth Conditions for ZZUPF95

Effects of inorganic salt content, carbon and nitrogen source types and content on growth and acid production capacity of ZZUPF95 were shown in Figure 1. Figure 1a,b has shown that the optimum inorganic salt content was 1.47%, which had the highest OD_600_, and there was no significant difference in acid production capacity among different concentrations of inorganic salt. After 24 h culture with different carbon sources, OD_600_ (Figure 1a) and pH (Figure 1b) of bacterial suspension were detected. OD_600_ in the saccharose group was significantly higher than that in other groups (*p* < 0.05), but the pH in the glucose and maltose groups was significantly lower than for other groups (*p* < 0.05). The effect of the carbon source on the growth of ZZUPF95 and its market price was evaluated, and glucose was selected as the optimum carbon source. Figure 1e,f shows that the OD600 in the 2% glucose was slightly higher and that there was a lower pH in all groups; there was no significant difference, so 2% glucose was selected as optimum carbon source. Using the same method, the optimum nitrogen source was 2% peptone (Figure 1g–j).

### 3.2. Determination of Optimal Growth Conditions for ZZUPF95

In RSM tests, 17 tests were conducted to estimate the pure error sum of squares. Results of the RSM analysis suggest that three independent variables were related as identified using the second-order polynomial equation: OD = 0.69–0.051 × A+0.040 × B+ 0.060 × C−0.019 × AB−0.019 × AC+0.027 × BC−011 × A^2–^0.023 × B^2^−0.029 × C^2^. The response surfaces for the impact of the independent variables on the average extraction efficiency of OD_600_ are shown in Figure 2a–c, which displayed the effect of the interaction of different glucose, peptone and inorganic salt contents on OD_600_, respectively. The contour diagram reflects the influence of the two variables on the dependent variable, the closer to the red area, the greater the value of the dependent variable OD_600_. According to the calculation results of the response surface, 1% glucose, 2% peptone and 1.47% inorganic salt were the optimal conditions, and OD_600_ was the highest. The ANOVA analysis table was shown in Table 1.

Figure 2d–f shows relationships between OD_600_ of ZZUPF95 bacterial suspension and incubation temperature, time and pH, respectively, and the optimal culture conditions of ZZUPF95 was at pH 7.0, in 37 °C for 48 h.

### 3.3. Optimization of FSBM Conditions for ZZUPF95

From Figure 3, the highest protease activity was achieved when the fermentation time was 36 h, the inoculation amount was 12.5%, and the ratio of feed to water was 1:1. K1, K2 and K3 in Table 2 are the sum of index of factor level 1, 2 and 3 respectively. Range R represents the influence of various factors in the test on the experimental index; the larger the R value, the higher the significance of the corresponding factor and the greater the influence on the dependent variable. The maximum value of K in each factor constitutes the optimal horizontal combination, so A1, B1 and C3 were selected out. However, since the scheme of this group was not in the orthogonal table, a validation experiment was carried out for this group, and it was verified that the acid protease activity of this group was 191 and greater than 171.05; therefore, A1B1C3 (fermentation time 36 h, inoculation amount 10%, ratio of material to water 1:1) is the optimal fermentation scheme.

### 3.4. Fermentation Quality, Chemical Composition and Microbial Population of FSBM

#### 3.4.1. Sensory Evaluation

After 3 d and 30 d fermentation, ZZUPF95 and ZZUPF95 + Protease groups always maintained more fluffy texture and had stronger sour aroma compared with other groups; while the color of Protease and ZZUPF95 + Protease groups were brown and with a fluffy texture. After 3 d of fermentation followed by 7 d aerobic exposure, CK had a strong foul smell and condensed into a mass; only CK was agglomerated after 30 d and 30–7 d. 

#### 3.4.2. Fermentation Quality of FSBM

Changes of pH in CK and different treatment groups during the 30 d fermentation period and aerobic exposure are presented in Figure 4a,b. The ability of reducing pH in different groups varied, for ZZUPF95 + Protease, ZZUPF95, Protease and CK to reduce pH below 5.00 the time required was 24 h (4.98), 36 h (4.77), 48 h (4.94) and 7 d (4.65), respectively. The lowest pH value during the whole fermentation and aerobic exposure periods was in the ZZUPF95 group, which reduced to 4.37 after 18 d of fermentation, and until the end of the aerobic exposure for 30–3 d.

As Figure 4c,d shows, LA content in ZZUPF95 + Protease group was significantly higher than that in other groups at 24 h and 18 d of fermentation (*p* < 0.05), which were 36.63 and 110.88 mg/g DM, respectively, and the other three treated groups were also significantly higher than that in CK at 36 h and 3 d of fermentation (*p* < 0.05); by 30 d, however, there was no significant difference among groups (*p >* 0.05). While in the aerobic exposure period, LA showed a significant difference in each group at 3–7 d (*p* < 0.05), among which ZZUPF95 with 88.04 and ZZUPF95 + Protease with 97.66 were significantly higher than that in CK 47.39 and Protease 59.20 mg/g DM (*p* < 0.05), and same results were observed at exposure after 30 d (*p* < 0.05). AA displays in Figure 4e,f indicated ZZUPF95 + Protease was the highest at 60 h of fermentation (*p* < 0.05), and there was also no significant difference in AA at 30 d as LA (*p >* 0.05). At 3–7 d, the AA content in ZZUPF95 + Protease reached 15.16 mg/g DM, which was significantly greater than that in other groups (*p* < 0.05).

#### 3.4.3. Chemical Composition of FSBM

The effect of fermentation on the nutritional changes in CP, EE, and CF are presented in Figure 5. From Figure 5a,b, it can be observed that the CP content of ZZUPF95 was at the highest level in all times of the fermentation periods; among them, at 36 h to 12 d of fermentation, the difference was significant compared with the other three groups; the same results also appeared at aerobic exposure except for fermentation of 30 d and then exposure of 7 d. Towards the EE content (Figure 5c,d), ZZUPF95 at 24, 60 h and 18 d, ZZUPF95 + Protease at 36, 48 h and the Protease at 30 d significantly increased the EE (*p* < 0.05); in three days of aerobic exposure, the EE of ZZUPF95 group was the highest, at 30–3 d, ZZUPF95 and ZZUPF95 + Protease were higher than others, while at 30–7 d, Protease and ZZUPF95 + Protease were superior. As shown in Figure 5e,f, the CF content of ZZUPF95 group and ZZUPF95 + Protease was significantly reduced at 36, 48 h, 7 and 12 d of fermentation; In aerobic exposure, the CF content in ZZUPF95 and ZZUPF95 + Protease group were also lower than the CK and Protease groups.

#### 3.4.4. Microbial Population of FSBM

The analysis of various microorganisms in the FSBM are demonstrated in Figure 6. In Figure 6a it was observed that the species and number of microorganisms in each group gradually decreased after 12 d of fermentation, especially in the ZZUPF95 group. Neither in the ZZUPF95 group from 48 h to 30 d, nor in the ZZUPF95 + Protease at 36, 48, 60 h and 18 d of fermentation, was coliform bacteria found. The clostridium was only observed in CK after 30 d fermentation. The microbial population in the exposure stage is shown in Figure 6b, which was less after fermentation of 30 d than those of 3 d, and the ZZUPF95 group still held a significant inhibitory effect on coliform bacteria (*p* < 0.05) that was not observed except after 3 d exposure after 3 d of fermentation. Furthermore, clostridium was only found in the CK and Protease groups during all aerobic exposure.

#### 3.4.5. SEM

Morphological features of SBM and FSBM were observed under a 2000 × electron microscope in Figure 7. FSBM after 3 d fermentation was more fragmentized, especially more pores and fragmented structure were found in Protease and ZZUPF95 + Protease groups, whereas SBM had a relatively smooth surface and the lamellar structure is relatively denser and complete. A big fragment was only observed in CK, while more loose networks with diffuse and big holes were detected in the treated groups after 30 d of fermentation.

#### 3.4.6. SDS-PAGE Profile

The protein patterns of FSBM are shown in Figure 8. It can be seen that after fermentation, the electrophoresis pattern of the SBM protein had changed. Protease and ZZUPF95 + Protease groups held the more significant effect on the degradation of large protein molecules, which was degraded to 20 and 30 KDa, while CK and ZZUPF95 has no difference; moreover, as opposed to the fermentation stage, the degradation effect was more obvious in aerobic exposure of ZZUPF95 + Protease group (*p* < 0.05). 

### 3.5. Microbial Community Dynamics of FSBM

#### 3.5.1. Alpha Diversity Indices of Microbial Community

The dynamics of the full microbiome, including the richness and diversity of the microbial community, were characterized by the alpha diversity shown as OTUS, Chao and Shannon indexes. In Table 3, the average Good’s coverage values of both bacterial and fungi for all FSBM were approximately greater than 97%. After fermentation and exposure treatment, the Chao and OTUS indexes of the ZZUPF95 and ZZUPF95 + Protease groups decreased compared with CK, and the Chao index of ZZUPF95 + Protease reached the minimum values of 1761.73 at 3–7 d; moreover, compared with other groups, Shannon and Simpson indexes in the ZZUPF95 + Protease group decreased most at each fermentation time and reached the lowest value also at 3–7 d, which was 2.37 and 0.65, respectively.

In fungi sequencing, except for 3 d of fermentation, the Chao and OTU indexes of ZZUPF95 and ZZUPF95 + the Protease group were increased; among them, the Chao index of the ZZUPF95 + Protease group and OTU of ZZUPF95 group reached the maximum at 30–7 d, which were 346.15 and 399.00, respectively. 

#### 3.5.2. Principal Coordinate Analysis

As shown in Figure 9, the result of principal coordinate analysis based on UniFrac (unweighted) distances clearly reflected the variance of the microbial community. For bacteria, it can be seen that CK were significantly separated from those in other groups at 3 d and 30 d of fermentation (*p* < 0.05) (Figure 9a,b). However, with exposure at 3–7 d, the intersection of ZZUPF95 and ZZUPF95 + Protease appeared, while at 30–7 d, CK and ZZUPF95 + Protease groups were more similar(Figure 9c,d). In the fungi community analysis (Figure 9e–h), CK of 3 d (Figure 9e) were also significantly separated from those in the other groups (*p* < 0.05) as bacteria, but the groups were overlapped after 30 d of fermentation and subsequent exposure.

#### 3.5.3. Abundance of Microbial Community

The abundance of the microbial community at phylum and genus levels was characterized and analyzed, with results being presented in Figure 10, which contains bacterial and fungi communities. For bacteria, at the phylum level (Figure 10a), the dominant microorganism of SBM was *Cyanobacteria* (93.61%); while after 3 d fermentation, the number of *Firmicutes* increased and dominated the fermentation among treated groups, *Firmicutes* in CK was 37.43, while in ZZUPF95, Protease and ZZUPF95 + Protease were 99.30, 98.97 and 98.86%, respectively; *Proteobacteria* was 58.82% in CK. After aerobic exposure for 7 d, *Firmicutes* in CK was 75.43, while treated groups have no significant changes. The phylum microflora structure of 30 d fermentation and aerobic exposure was similar to 3 d aerobic exposure. At the genus level (Figure 10b), after 3 d fermentation, bacterial flora structure had changed in four groups, *Enterococcus* in CK was 16.87%, was the dominant bacterial, while *Weissella* was dominant in ZZUPF95 and Protease, at 38.51 and 62.51%, respectively, *Pediococcus* with 47.11 became the dominant bacterial in ZZUPF95 + Protease. From 3–7 d, the higher abundance of *Lactobacillus* was found in ZZUPF95 and ZZUPF95 + Protease, which were 27.36 and 25.41%, respectively. *Enterococcus* in CK, ZZUPF95 and ZZUPF95 + Protease increased to 28.58, 57.56 and 61.48%, respectively, while *Weissella* increased to 19.76 in CK and reduced to 50.17% in Protease, *Pediococcus* in ZZUPF95 + Protease from 47.11 to 0.47% after aerobic exposure 7 d. After 30 d fermentation, the relative abundance of *Lactobacillus* increased in the treated groups compared to 3 d fermentation, and the dominant bacterial in CK was *Enterococcus* with 37.86% relative abundance, in ZZUPF95 and Protease were *Weissella* with 37.17 and 37.00% relative abundance, in ZZUPF95 + Protease was *Pediococcus* and *Enterococcus*. There were no significant changes in the microbiota structure in each group after 7 d aerobic exposure.

As for fungi, at the phylum level (Figure 10c), *Ascomycota* was the dominant phylum in all groups, and had no significant changes after 3, 30 d fermentation and 7 d aerobic exposure at the genus level (Figure 10d). After 3 d fermentation, *Fusarium* were higher in CK and ZZUPF95 + Protease groups with 9.99 and 24.17%, respectively, and *Kazachstania* was 13.61% in ZZUPF95 + Protease. With aerobic exposure for 7 d, *Aspergillus* in CK increased from 4.45 to 24.78%, and *Kazachstania* increased from 2.51 to 10.67%. After 30 d fermentation, *Fusarium* reduced significantly in CK (3.43%) and ZZUPF95 + Protease (3.65%) compared to 3 d fermentation, and lower *Kazachstania* (3.90%) in ZZUPF95 + Protease, while higher *Kazachstania* (9.37%) in *P*. *Fusarium*, *Aspergillus* and *Kazachstania* higher in CK and ZZUPF95 compared with ZZUPF95 + Protease after 7 d aerobic exposure.

### 3.6. Correlation Analyses of the Bacterial and Fungi Community with Fermentation Properties

The heatmap showing the Pearson correlation coefficients between FSBM and relative abundances of dominant microbial genera is presented in Figure 11. pH was positively correlated with *Lactococcus* from 3 d until the end of the experiment (*p* < 0.05), and negatively correlated with *Kazachstania*, *Fusarium*, *Cladosporium*, *Alternaria*, *Acremonium* and *Candida* after 3–7 d (*p* < 0.01) and 30–7 d (*p* < 0.05), respectively. As for organic acid, AA was positively correlated with *Kazachstania* from 3–7 d (*p* < 0.05).

## 4. Discussion

Heating, soaking, enzymatic hydrolysis and microbial solid-state fermentation are the usual ways of processing SBM to degrade the large molecules of protein and eliminate their anti-nutritional factors [20]. Of these, fermentation can improve the digestibility of agricultural by-products by livestock, and fermented feed contains beneficial microorganisms and has the function of probiotics [21]. In addition, after banning the use of antibiotics in animals, biological additives as a substitute for antibiotics have attracted even more unprecedented attention [22]. The fermentation process and the quality of the final feed are largely influenced by the characteristics of the raw material in general, and for high-quality fermented feed, the number of LAB should reach 5.0 lg cfu/g FM of material. Although the population of LAB on SBM in this study was 7.0 lg cfu/g FM, higher amounts of unwanted bacteria aerobic bacteria, coliform bacteria and bacilli were also observed with more than 4.5 lg cfu/g FM, which would result in poor fermentation without exotic additives [23].

LAB including *E. faecium*, *E. faecalis*, *Lactobacillus* (*L.*) *brevis*, *L. casei*, *L. plantarum* and *L. buchneri* are able to promote lactic acid production to reduce the pH; thereby benefitting the host and being widely used as additives for feed fermentation as SBM [24]. To maximize the effect of fermented feedstuffs, suitable fermentation conditions must be identified and optimized accordingly based on microorganism species and incubation temperature and time. Therefore, a protease producing *E. faecalis* ZZUPF95 with excellent growth performance was used for SBM fermentation in this research, and the optimum growth conditions and fermentation process of ZZUPF95 was studied first in order to get the best fermentation results. As inorganic salts, carbon and nitrogen sources are the basic components of the MRS medium, and the types and contents of these three substances were screened. According to the contour map and design-Expert software analysis, the best medium ratio of ZZUPF95 was glucose 1%, peptone 2% and inorganic salt 1.47%, and the best culture conditions were obtained on the basis of the optimum medium formula. Similarly, Choi et al. [25] obtained the optimal medium for *L. plantarum* 200,655 through RSM experiments on carbon and nitrogen sources. The fermentation condition is another key factor to the success of fermentation [26]. A fermentation time of 36 h, an inoculation amount of 10%, and a ratio of material to water of 1:1 is the optimal fermentation scheme. In the study of Sun et al. [27], they had investigated the fermentation conditions under which *L. brevis* secretes protease, and the optimized fermentation conditions were a fermentation time of 36 h, initial pH of 5.0 and fermentation temperature of 42 °C.

Sensory evaluation is the first step in judging the quality of the feed, which is mainly based on colour, smell and texture. As acid is a good flavor enhancer, and the ZZUPF95 and ZZUPF95 + Protease groups that had a strong, sour aroma would improve palatability considerably. CK, which had a strong foul smell, condensed into a mass at the aerobic exposure stage, which might have been because the environment changed to enable aerobic fungi to proliferate substantially. The process of fungi growth produces toxins that can seriously damage animals, while the sensory evaluation shows that the addition of LAB avoids the possible harm caused by fungi. 

The appropriate pH for the growth of several pathogenic bacteria is neutral to alkaline, e.g., *Escherichia coli*, *Staphylococcus* and *Clostridia* are 6.0–8.0, 6.8–7.5 and 6.0–7.5, respectively, and when the pH is less than 4.0, they are inactivated in large numbers. Therefore, by lowering the pH, organic acids inhibit the multiplication of harmful microorganisms and reduce the consumption of nutrients, while promoting the proliferation of beneficial bacteria [28,29]. By comparing the pH changes of FSBM in this research, the pH of ZZUPF95 and ZZUPF95 + Protease groups decreased faster in fermentation, and the ZZUPF95 group even had the lowest pH in the aerobic exposure stage, which may be because LAB exerts an antibacterial effect, slows down the proliferation of aerobic bacteria, and thus prevents the increase of pH. Organic acids such as lactic acid and acetic acid can lower the pH in the fermentation environment and inhibit bacteria by disrupting bacterial cell membranes, interfering in bacterial enzyme synthesis, affecting bacterial DNA replication, and improves the utilization of minerals and nutrients, which are generally used to assess the quality of fermented feedstuffs [30,31,32]. Lactic acid, a secondary metabolite of the LAB, which is one of the representative organic acids of fermented feedstuffs, and plays an important role in the defense mechanism and inhibits pathogenic bacteria, thus promoting the rapid colonization of LAB [33]. In the present study, regardless of vacuum fermentation or aerobic exposure, LAB added groups had the most significant increase in lactic acid content, which are positively correlated with pH reduction and broad-spectrum bacteriostatic activity. Similarly, Saelim et al. found that the antibacterial activity of *L. plantarum* S0/7 isolated from fermented smelly bean was mainly due to the production of organic acids [34]. The mixed group had a significant effect on increasing the content of acetic acid, which can also inhibit the growth of pathogenic bacteria and improve the utilization of minerals and nutrients [35]. 

Changes in CP are one of the most important parameters in FSBM and it is often hydrolysed into peptides, amino acids and ammonia by protease and microbial activity [36,37]. In chemical composition analysis, fermentation and aerobic exposed processes with LAB significantly increased CP and EE contents, while decreasing CF content, respectively. The increase in CP content of FSBM comes at the expense of the concentration of the SBM components, as LAB consume organic matter through respiration, releasing carbon dioxide and water, reducing the total amount of product during the fermentation process. For the significant increase in CP after 30 d of fermentation, this may be due to the production of bacteriophage proteins, where the microorganisms use the non-protein nitrogen in the SBM as well as nitrogenous anti-nutritional factors as nutrients and synthesise bacteriophage proteins, which also increases the nutritional value of the FSBM [38].

Compared with SBM and CK, all treated groups showed looser networks and bigger holes structures. These changes indicate that microbial activity has begun to degrade the soy protein, producing small molecules, peptides and various amino acids. The accumulation of lactic acid and the protease activity declined at 30 d of fermentation and led to an increase in the acidity of the environment, resulting in the development of soy protein aggregation, which manifests itself in the form of even more severe damage to the protein surface structure.

Protease and ZZUPF95 + Protease groups held the more significant effect on the degradation of large protein molecules compared with that of SBM, CK and ZZUPF95, and the degradation effect was more obvious in the aerobic exposure of ZZUPF95 + Protease group. The mixed group fermented SBM by LAB with complementary protease to synergistically degrade large molecule proteins in feed ingredients into small molecule peptides that can be easily absorbed by animals, thereby increasing the nutritional value and conversion rate of feed [39]. 

The addition of LAB can obtain faster accumulation of lactic acid and lower pH values, and inhibit the growth of undesirable bacteria [40]; thus, the LAB count was higher in the ZZUPF95 group than that in the CK and in the mixed at the vast majority of the fermentation time, which was consistent with the decrease in pH at the same stage; meanwhile, coliform bacteria was inhibited in these two groups. Rapid proliferation of yeast and mold consumes large amounts of nutrients and caused a pH increase during the aerobic exposure phase and led FSBM into eventual deterioration. *Clostridium*, whose fermentation products are mainly butyric acid and NH_3_-N that produce unpleasant odors, was only detected in the CK and mixed groups during the aerobic stage, and this was in line with the sensory results [41,42].

The decrease in OTU number, Shannon and Chao indexes with the addition of LAB, suggested that *E. faecalis* ZZUPF95 inhibited other bacteria and fungi and became the dominant bacterium. In principal coordinate analysis, as fermentation progressed, the dispersion of the bacterial community was greater in each treatment group, which suggested that fermentation time affects bacterial communities. At 3–7 d, the overlap between ZZUPF95 and the mixed group indicated that the two groups had similar microbial communities. 

The dominant bacteria of SBM was a Gram-negative bacterium, *Cyanobacteria*, which has many types that are identified as toxin producers; as fermentation progressed, fermentation resulted in *Firmicutes*, a Gram-positive phylum that produce acid and a variety of enzymes and grows easily in the acidic and anaerobic environment of the sealed fermentation process, and it became the predominant community in all groups, while after 3 d until the end of the whole fermentation only with relative abundance differed from each other, even including all aerobic exposure stages. *Firmicutes* is one of the most common phylum among the gut microbiota, and has many beneficial bacteria members such as *Enterococcus* and *L**actobacillus* [43]. The bacterial community during the fermentation of SBM showed a dynamic change from Gram-negative to positive, indicating that the presence of abundant pathogenic bacteria in the raw material that replaced LAB that is beneficial to feed became the dominant bacteria after fermentation. At the genus level, both at 3 d and 30 d fermentation, the dominant bacterial species in CK was *Enterococcus*, and in ZZUPF95 and protease it was *Weissella*, while the difference was the abundance after 30 d was higher than that of 3 d; for ZZUPF95 + the Protease group, at 3 d it was *Pediococcus* and at 30 d it was *Pediococcus* and *Enterococcus*. That is, the dominant bacteria in all groups were LAB, which may be caused by the continuous decrease of the pH of fermentation system inhibiting the growth of acid sensitivity bacteria, while LAB could grow and reproduce normally, and this is also the main reason for the significant improvement in fermentation quality of FSBM. 

*Ascomycota*, which is the most diverse and species-rich phylum in kingdom fungi and shows a broad range of life modes such as pathogenic (agriculturally and clinically), saprobic and endophytic [44], was the dominant phylum in all groups at the phylum level after 3 d and 30 d fermentation and 7 d aerobic exposure. At the genus level, after 3 d fermentation, *Fusarium*, which with numerous species are important pathogens, can produce ranges of secondary metabolites and cause opportunistic mycoses [45], were higher in the CK and ZZUPF95 + Protease groups; *Kazachstania* could assimilate lactic acid and hydrolyzes glucuronide as a metabolic substrate for LAB, facilitating its use of fructose to produce acetic acid, and it was also higher in the ZZUPF95 + Protease group [46]. This is also consistent with the aforementioned ZZUPF95 + Protease group, which had a significant effect on increasing the content of acetic acid; in additional, acetic acid was positively correlated with *Kazachstania*. *Aspergillus*, many species of which are pathogenic bacteria capable of producing toxins, increased nearly sixfold in CK after aerobic exposure for 7 d. After 30 d fermentation, *Fusarium* reduced significantly in the CK and ZZUPF95 + Protease groups compared to that at 3 d. *Cladosporium*, a dematiaceous mold widely distributed in air and rotten organic material and frequently isolated as a contaminant on foods, *Alternaria* which can cause plant and animal diseases, *Acremonium*, the metabolites of which are trichothecene crotocin and cerulenin and can enhance aflatoxin biosynthesis [47], and *Candida,* which poses a risk to animal health in the form of opportunistic infections, were all negative with pH; in other words, they were inhibited in the acidic environment during the aerobic exposure phase. 

The artificial addition of inoculated microorganisms not only increased the number of added microorganisms but promoted the formation of symbiosis between some other functional microorganisms and inoculated microorganisms [48]. In this study, *Enterococcus* was positively correlated with LA and negatively with pH value, meanwhile LA was negatively correlated with *Pediococcus*, *Weissella*, *Acetobacter* and *Bacillus*, which were out-competed by *Enterococcus* at low pH condition and inhibited because a high LA concentration was observed after 3 d fermentation. However, LA was positively correlated with some yeast species the at aerobic exposure stage, which may be because they are acid-assimilating yeasts that consume organic acid to increase the pH [41].

## 5. Conclusions

Glucose 2%, peptone 2% and inorganic salt 1.47% is the best medium ratio of *E. faecalis* ZZUPF95, and the optimal fermentation scheme is a fermentation time of 36 h, an inoculation amount of 10%, and a ratio of material to water of 1:1. Compared to the CK and protease treated groups of FSBM, the LAB addition group showed reduced pH, the inhibition of pathogenic bacteria and the degradation of CF, especially against coliform bacteria, while LA, CP and EE were increased. And the addition of the protease group had an effect on the degradation of macromolecular protein. The microbial community composition with the addition of ZZUPF95 and the extension of fermentation time resulted in the microbial diversity decreasing significantly, and the abundance of bacteria and fungi alsochanged. Therefore, *E. faecalis* ZZUPF95 might be considered as potential feed additives to improve the quality of FSBM, especially ZZUPF95 + Protease with protease.

## Figures and Tables

**Figure 1 microorganisms-10-00530-f001:**
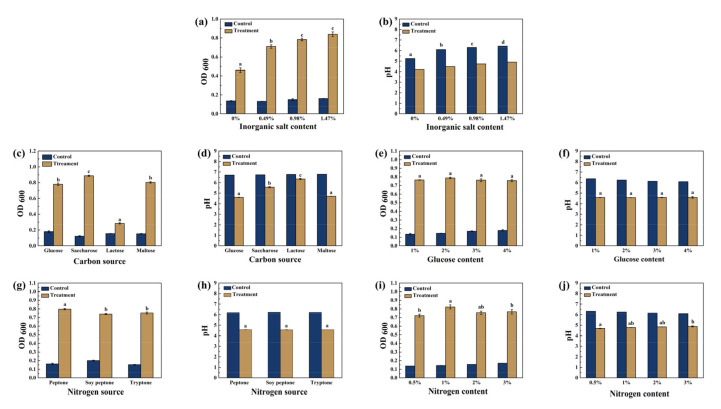
Effects of inorganic salt content, carbon and nitrogen source types and content on growth and acid production capacity of ZZUPF95. (**a**,**b**), inorganic salt content; (**c**,**d**), carbon source types; (**e**,**f**), carbon content; (**g**,**h**), nitrogen source types; (**i**,**j**), nitrogen content. Different lowercase letters (a–d) for the same treatment time indicate significant differences (*p* < 0.05) among different treatment groups.

**Figure 2 microorganisms-10-00530-f002:**
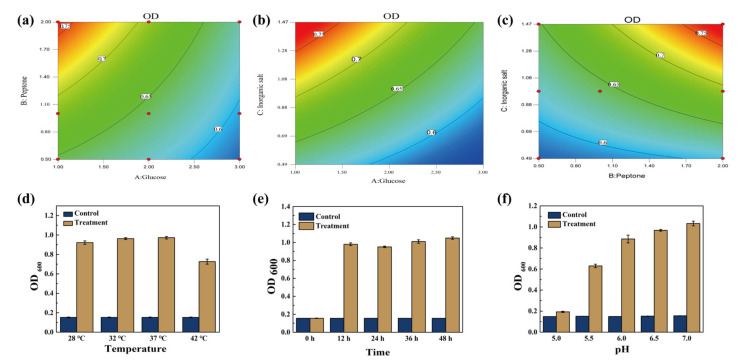
Optimal glucose, peptone, inorganic salt, temperature, time and pH for growth of ZZUPF95. (**a**–**c**), contour plots for growth of ZZUPF95 under the influence of glucose, peptone and inorganic salt contents; (**d**–**f**), effects of temperature, time and pH on growth of ZZUPF95, respectively.

**Figure 3 microorganisms-10-00530-f003:**
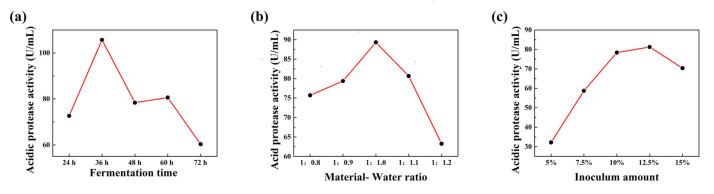
Relationship between protease activity and fermentation time, material-water ratio and inoculum amount. (**a**), fermentation time; (**b**), material- water ratio; (**c**), inoculum amount.

**Figure 4 microorganisms-10-00530-f004:**
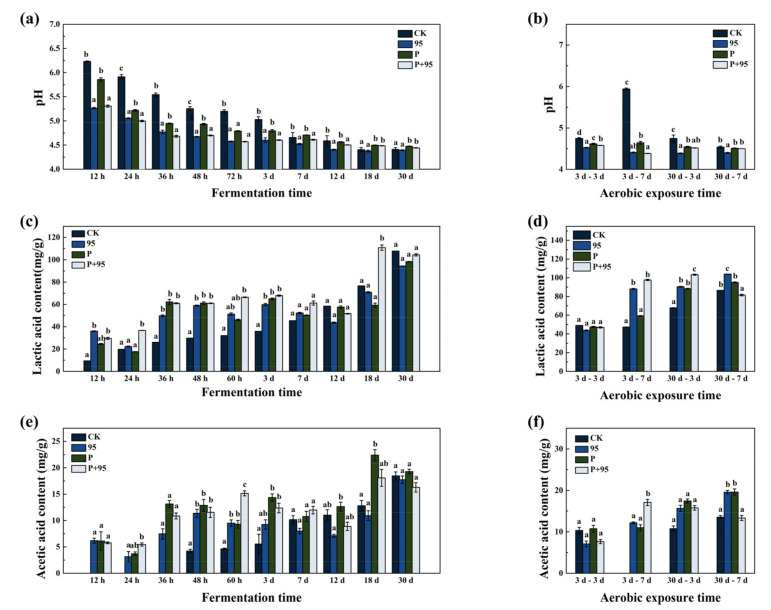
Fermentation quality of FSBM during fermentation and aerobic exposure time. (**a**,**b**), pH; (**c**,**d**), lactic acid; (**e**,**f**), acetic acid. Different lowercase letters (a–d) for the same treatment time indicate significant differences (*p* < 0.05). 95, ZZUPF95; P, Protease; P + 95, ZZUPF95 + Protease; d, days.

**Figure 5 microorganisms-10-00530-f005:**
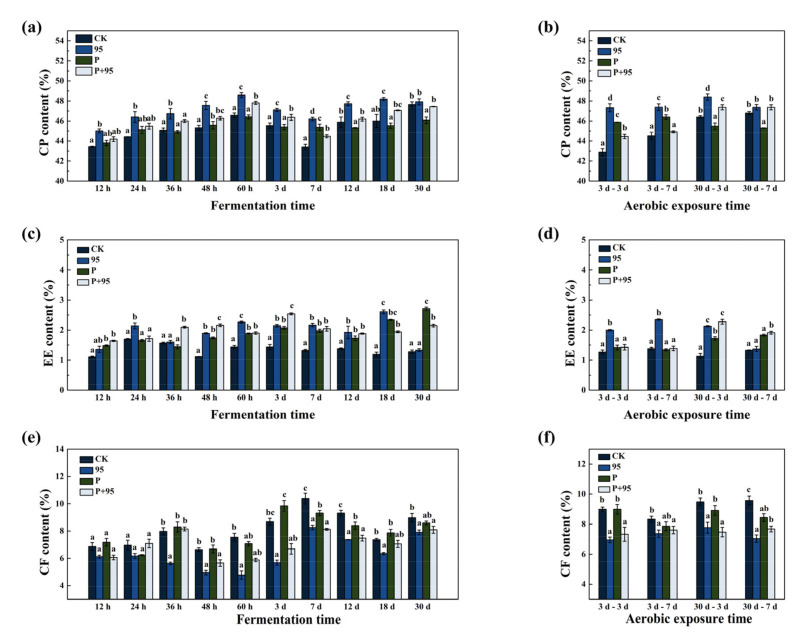
Chemical composition of FSBM during fermentation and aerobic exposure. (**a**,**b**), CP, crude protein; (**c**,**d**), EE, ether extract; (**e**,**f**), CF, crude fiber; DM, dry matter. Different lowercase letters (a–d) for the same treatment time indicate significant differences (*p* < 0.05). 95, ZZUPF95; P, Protease; P + 95, ZZUPF95 + Protease; d, days.

**Figure 6 microorganisms-10-00530-f006:**
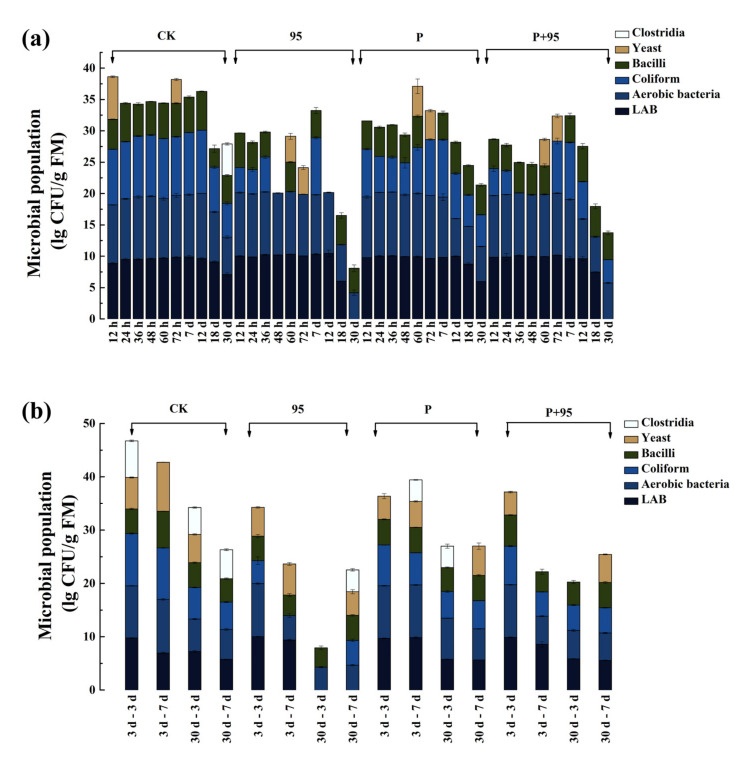
Microbial population of FSBM during fermentation and aerobic exposure. (**a**), fermented from 12 h to 30 d; (**b**) aerobic exposure time. 95, ZZUPF95; P, Protease; P + 95, ZZUPF95 + Protease; d, days.

**Figure 7 microorganisms-10-00530-f007:**
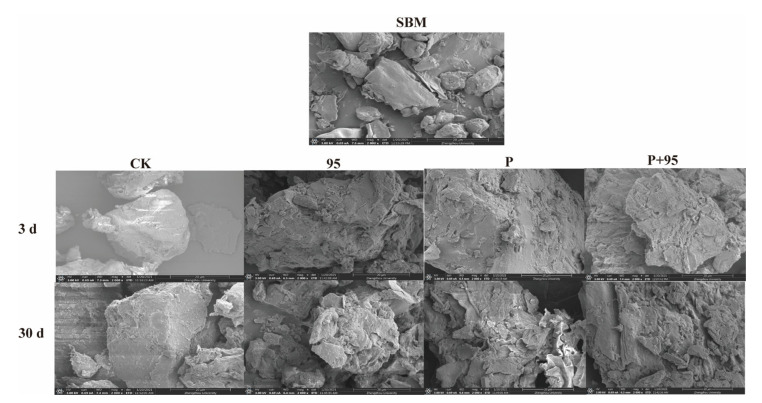
Surface structure of SBM and FSBM after 3 and 30 d of fermentation in 4 treatments. SEM images at ×2000-fold magnification. 95, ZZUPF95; P, Protease; P + 95, ZZUPF95 + Protease; d, days.

**Figure 8 microorganisms-10-00530-f008:**
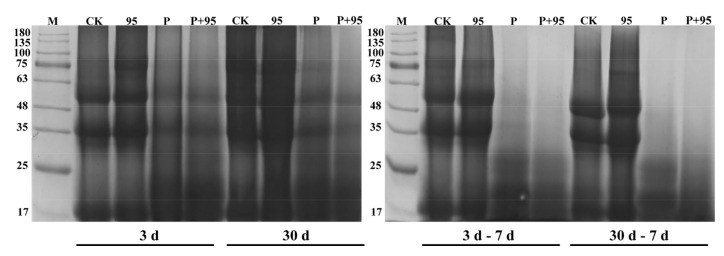
Protein degradation after fermentation and aerobic exposure. M, marker; 95, ZZUPF95; P, Protease; P + 95, ZZUPF95 + Protease; d, days.

**Figure 9 microorganisms-10-00530-f009:**
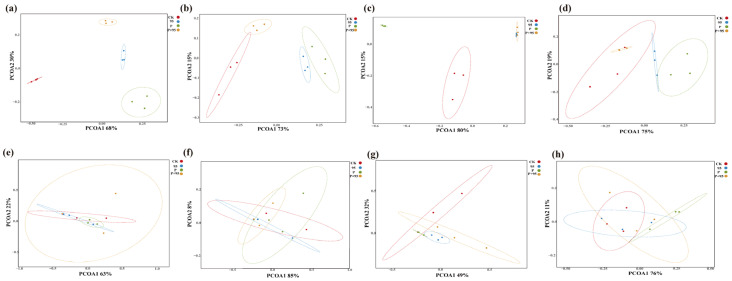
Principal coordinates analysis (PCoA) on the genus level of bacterial and fungi community during fermentation and aerobic exposure. (**a**–**d**): PCoA of bacterial community on 3 d, 30 d, 3–7 d and 30–7 d, respectively; (**e**–**h**): PCoA of fungi community on 3 d, 30 d, 3–7 d and 30–7 d, respectively. 95, ZZUPF95; P, Protease; P + 95, ZZUPF95 + Protease; d, days.

**Figure 10 microorganisms-10-00530-f010:**
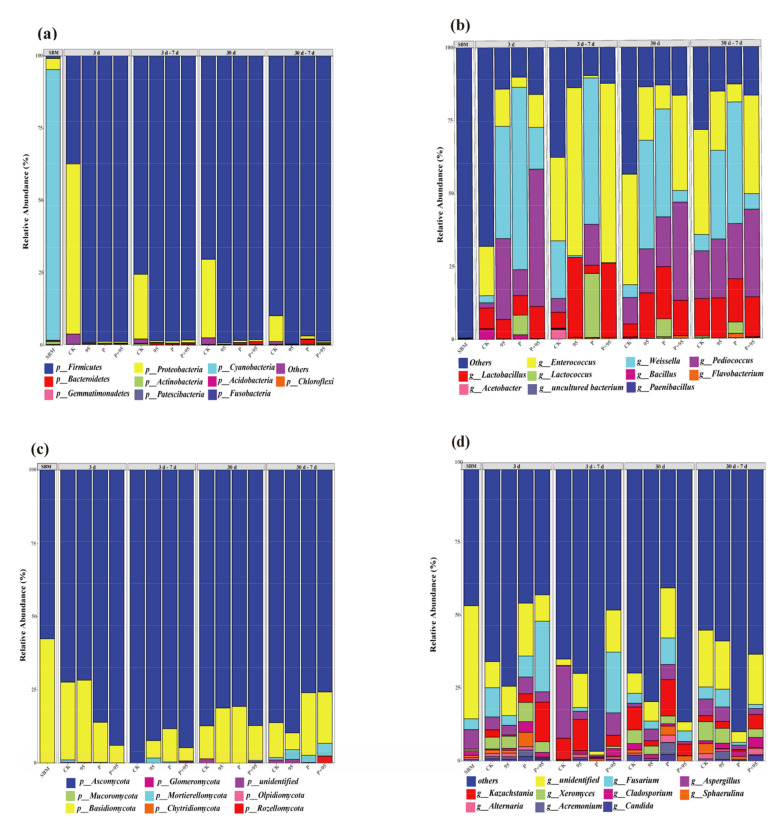
The bacterial and fungi community of FSBM during fermentation and aerobic exposure. The bacterial communities are shown at the phylum level (**a**) and the genus level (**b**). The fungi communities are shown at the phylum level (**c**) and the genus level (**d**). 95, ZZUPF95; P, Protease; P + 95, ZZUPF95 + Protease; d, days.

**Figure 11 microorganisms-10-00530-f011:**
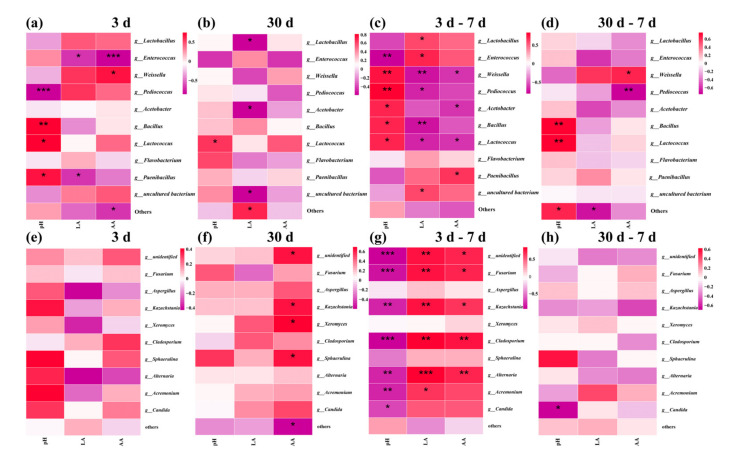
Spearman correlation heatmap of abundance of the top 10 enriched bacteria and fungi at the genus level with fermentation properties during fermentation and aerobic exposure. Bacterial: 3 d (**a**) and 30 d (**b**) of fermentation, 3–7 d (**c**) and 30–7 d (**d**) of aerobic exposure. Fungi: 3 d (**e**) and 30 d (**f**) of fermentation, 3–7 d (**g**) and 30–7 d (**h**) of aerobic exposure. *, *p* < 0.05; **, *p* < 0.01; ***, *p* < 0.001. d, days.

**Table 1 microorganisms-10-00530-t001:** Analysis of variance for the quadratic polynomial mode.

Source	Sum of Squares	DF	Mean Square	F Value	*p*-Value
Model	0.068	6	0.011	13.74	0.0003
A	0.023	1	0.023	28.39	0.0003
B	0.016	1	0.016	18.82	0.0015
C	0.027	1	0.027	33.04	0.0002
AB	1.53 × 10^−3^	1	1.53 × 10^−3^	1.85	0.2037
AC	1.44 × 10^−3^	1	1.44 × 10^−3^	1.75	0.2159
BC	3.04 × 10^−3^	1	3.04 × 10^−3^	3.68	0.0841
Residual	8.27 × 10^−3^	10	8.27 × 10^−4^		
Cor Total	0.076	16			
Std. Dev.	0.029	R-Squared	0.8918		
Mean	0.65	Adj R-Squared	0.827		
C.V. %	4.44	Pred R-Squared	0.6249		
PRESS	0.029	Adeq Precision	11.631		

Note: DF, degree of freedom; CV, coefficient of variation; A, glucose; B, peptone; Inorganic salt.

**Table 2 microorganisms-10-00530-t002:** The results of orthogonal test L_9_ (3^3^) for soybean meal fermentation with lactic acid bacteria liquid fermentation broth.

Treatment	Fermentation Time	Inoculum Mount	Material-Water Ratio	Acid Protease Activity
1	1	1	1	136.84
2	2	2	1	171.05
3	3	3	1	110.67
4	3	2	2	92.71
5	2	1	2	127.77
6	1	3	2	148.30
7	1	2	3	147.27
8	2	3	3	129.14
9	3	1	3	168.82
K1	144.13	144.50	139.52	
K2	142.65	137.01	122.93	
K3	124.07	129.37	148.41	
R	20.07	15.11	25.49	

**Table 3 microorganisms-10-00530-t003:** Alpha diversity of bacteria and fungi during fermentation and aerobic exposure.

Days	Treatments	Bacteria				Fungi			
		Shannon	Chao 1	OTUS	Good’s Coverage	Shannon	Chao 1	OTUs	Good’s Coverage
3 d	CK	5.83	3521.64	1785	0.99	4.03	99.67	115	0.98
	95	4.76	3009.98	1301	0.99	4.01	81.72	83	0.99
	P	4.78	3278.14	1545	0.99	5.01	225.69	321	0.97
	P + 95	4.32	2577.36	1244	0.99	3.45	91.07	103	0.99
3–7 d	CK	5.8	3516.52	1753	0.99	1.69	11.98	32	0.99
	95	2.6	1923.12	741	0.99	3.45	198.05	316	0.98
	P	4.11	3038.66	1461	0.99	1.01	37.93	58	0.99
	P + 95	2.37	1761.73	708	0.99	4.04	218.7	385	0.96
30 d	CK	5.41	3213.72	1456	0.99	3.97	63.65	83	0.99
	95	4.74	2062.2	1017	0.99	3.23	56.41	71	0.99
	P	5.03	2610.79	1265	0.99	5.87	355.33	356	0.98
	P + 95	3.9	1807.25	841	0.99	2.95	23.17	28	0.99
30–7 d	CK	5.2	2824.77	1332	0.99	4.82	293.78	361	0.98
	95	4.75	2340.07	993	0.99	5.34	290.42	399	0.97
	P	4.89	3252.91	1571	0.99	4.26	79.27	97	0.99
	P + 95	3.99	2469.96	1064	0.99	4.2	346.15	369	0.98

Note: 95, ZZUPF95; P, Protease; P + 95, ZZUPF95 + Protease; d, days.

## Data Availability

Data are contained within the article.

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
