# Peer review of "Microbial Population Succession and Community Diversity and Its Correlation with Fermentation Quality in Soybean Meal Treated with Enterococcus faecalis during Fermentation and Aerobic Exposure"

_microorganisms, 2022, doi:10.3390/microorganisms10030530_

Round 1
Reviewer 1 Report
Fermentation is one of the most promising options for degrading the macromolecular proteins in soybean meal, removing its anti-nutritional factors and improving its nutritional value. But few studies on the microbial dynamic changes and fermentation quality of fermented soybean meal. This is a very interesting paper aimed to evaluate the effects of a protease-producing Enterococcus faecalis on fermentation characteristics and microbial communities during ensiling and aerobic exposure phases of soybean meal. The results of the study will provide a scientific basis for revealing the influence mechanism of lactic acid bacteria on microbial changes, fermentation quality and aerobic stability during the fermentation of soybean meal. The manuscript is very well-written, so the submission is worthy of publication. In order to be considered for publication some minor revisions are necessary. 1. Line 20. Change Glucose 2%, Peptone 2% to glucose 2%, peptone 2%. 2. Line 34. SBM must be explained because it is the first time mentioned in introduction. 3. Line 72. DM (dry matter). 4. Line 90. optical density (OD) 5. Line 109. Please modify the order as follows: fermentation quality, chemical composition and microbiology. 6. Line 47, 122, 156. Please modify the font of the temperature. 7. Line 231. Please modify this sentence. 8. Line 322-324. 3 d fermentation, P and P+95 are not at × 2000-fold magnification, Please check and redraw. 9. Line 339-340. Please add the average Good’s coverage values. 10. Line 431. This paragraph is not referenced. Who said that? 11. Line 447. glucose 1% or 2%, Please check. 12. Line 495-498. This paragraph is not referenced. 13. Line 555-557. Please add references.Author Response
Reviewer 1
Comments and Suggestions for Authors
Point 1: Line 20. Change Glucose 2%, Peptone 2% to glucose 2%, peptone 2%.
We are so sorry that due to our negligence there were something went wrong, thank you for reminding us, and we have revised in Line 23.
Point 2: Line 34. SBM must be explained because it is the first time mentioned in introduction.
Thank you for the reminder, SBM is short for soybean meal, and we have added the information to the manuscript, please see Line 37.
Point 3: Line 72. DM (dry matter).
Thank you for reminding us, and we have revised in Line 74, 75.
Point 4: Line 90. optical density (OD)
Thank you, we have increased the optical density in Line 92.
Point 5: Line 109. Please modify the order as follows: fermentation quality, chemical composition and microbiology
Thank you. We have adjusted the order of fermentation quality, chemical composition and microbiology in Line 111.
Point 6: Line 47, 122, 156. Please modify the font of the temperature.
Thank you. We have revised the font of the temperature, please see Line 49, 130 and 167.
Point 7: Line 231. Please modify this sentence.
Thank you. We have revised this sentence as K1, K2 and K3 in Table.2, are the sum of index of factor level 1, 2 and 3 respectively. Please see Line 254.
Point 8: Line 322-324. 3 d fermentation, P and P+95 are not at × 2000-fold magnification. Please check and redraw.
Thank you, we have added the SEM of P and P+95 at × 2000-fold magnification and redrawn the image as following. Please see Figure 7 .
Figure 7. Surface structure of SBM and FSBM after 3 and 30 d of fermentation in 4 treatments. SEM images at × 2000-fold magnification. 95, ZZUPF95; P, Protease; P+95, ZZUPF95+Protease; d, days.
Point 9: Line 339-340. Please add the average Good’s coverage values.
Thanks for the good suggestion. We have added Good’s coverage values in Table 3.
Days |
Treatments |
Bacteria |
Fungi |
||||||
Shannon |
Chao1 |
OTUS |
Good's coverage |
Shannon |
Chao 1 |
OTUs |
Good’s coverage |
||
3 d |
CK |
5.83 |
3521.64 |
1785 |
0.99 |
4.03 |
99.67 |
115 |
0.98 |
95 |
4.76 |
3009.98 |
1301 |
0.99 |
4.01 |
81.72 |
83 |
0.99 |
|
P |
4.78 |
3278.14 |
1545 |
0.99 |
5.01 |
225.69 |
321 |
0.97 |
|
P+95 |
4.32 |
2577.36 |
1244 |
0.99 |
3.45 |
91.07 |
103 |
0.99 |
|
3 d - 7 d |
CK |
5.8 |
3516.52 |
1753 |
0.99 |
1.69 |
11.98 |
32 |
0.99 |
95 |
2.6 |
1923.12 |
741 |
0.99 |
3.45 |
198.05 |
316 |
0.98 |
|
P |
4.11 |
3038.66 |
1461 |
0.99 |
1.01 |
37.93 |
58 |
0.99 |
|
P+95 |
2.37 |
1761.73 |
708 |
0.99 |
4.04 |
218.7 |
385 |
0.96 |
|
30 d |
CK |
5.41 |
3213.72 |
1456 |
0.99 |
3.97 |
63.65 |
83 |
0.99 |
95 |
4.74 |
2062.2 |
1017 |
0.99 |
3.23 |
56.41 |
71 |
0.99 |
|
P |
5.03 |
2610.79 |
1265 |
0.99 |
5.87 |
355.33 |
356 |
0.98 |
|
P+95 |
3.9 |
1807.25 |
841 |
0.99 |
2.95 |
23.17 |
28 |
0.99 |
|
30 d - 7 d |
CK |
5.2 |
2824.77 |
1332 |
0.99 |
4.82 |
293.78 |
361 |
0.98 |
95 |
4.75 |
2340.07 |
993 |
0.99 |
5.34 |
290.42 |
399 |
0.97 |
|
P |
4.89 |
3252.91 |
1571 |
0.99 |
4.26 |
79.27 |
97 |
0.99 |
|
P+95 |
3.99 |
2469.96 |
1064 |
0.99 |
4.2 |
346.15 |
369 |
0.98 |
Table. 3 Alpha diversity of bacteria and fungi during fermentation and aerobic exposure
Point 10: Line 431. This paragraph is not referenced. Who said that?
Thank you, we have increased the corresponding reference as following into manuscript.
Cai Y.; Benno Y.; Ogawa M.; Kumai, S. Effect of applying lactic acid bacteria isolated from forage crops on fermentation characteristics and aerobic deterioration of silage. J. Dairy Sci. 1999, 82(3):520-526. doi: 10.3168/jds.S0022-0302(99)75263-X.
Point 11: Line 447. glucose 1% or 2%, Please check.
We are so sorry that due to our negligence there were something went wrong, thank you for reminding us, and we have revised in the whole manuscript, the correct glucose level is 1%.
Point 12: Line 495-498. This paragraph is not referenced.
Thank you, we have added the corresponding reference into the manuscript.
Jo, H.; Kim, E.B.; Han, G.G.; Kim, B.G. The Influence of dietary supplementation of bacteriophages on energy and nutrient digestibility and intestinal microbiota of pigs. FASEB J. 2017, 31. doi:10.1080/00071668.2014.991272.
Point 13: Line 555-557. Please add references.
Thank you, we have increased the corresponding reference into the manuscript.
Carbonetto, B.; Nidelet, T.; Guezenec, S.; Perez, M.; Segond, D.; Sicard, D. Interactions between Kazachstania humilis yeast species and lactic acid bacteria in sourdough. Microorganisms. 2020, 8(2). doi: 10.3390/microorganisms8020240.

Reviewer 2 Report
The manuscript entitled "Microbial population succession and community diversity and its correlation with fermentation quality in soybean meal treated with Enterococcus faecalis during fermentation and aerobic exposure" describes the optimization of the fermentation process of soybean meal as well as the characterization of the fermented product. It is well written, anyway, there are a few details that need to be clarified:
Line 20 - What is the meaning of CK?
Line 111 - How the color, smell and texture of SBM and FSBM were evaluated? please, specify.
line 211 - Please, add more information about the ANOVA analysis. Which parameters were significant? moreover, what was the R2, adjusted R2 and predicted R2, Adeq Precision and CV%?
Author Response
Reviewer 2
Comments and Suggestions for Authors
Point 1: Line 20 - What is the meaning of CK?
We are very sorry that we did not clarify the term CK in the manuscript, which may also bring some readers reading doubts. CK means “Control check”, which is a commonly used expression for control group in some research articles, such as the following references. CK in this research means “soybean meal without additives”, and we have added the explanation of CK in the manuscript, please see Line 22 and 103.
- Zhao, J.; Dong, Z.; Li, J.; Chen, L.; Bai, Y.; Jia, Y.; Shao, T. Ensiling as pretreatment of rice straw: The effect of hemicellulase and Lactobacillus plantarum on hemicellulose degradation and cellulose conversion. Bioresource Technol. 2018, 266, 158-165. doi:10.1016/j.biortech.2018.06.058.
- Yin, Y.; Pereira, J.; Zhou, L.; Lorenzo, J.M.; Tian, X.; Zhang, W. Insight into the effects of sous vide on cathepsin b and l activities, protein degradation and the ultrastructure of beef. Foods. 2020, 9, 1441. doi: 10.3390/foods9101441.
- Zhao, S.; Yang, F.; Wang, Y.; Fan, X.; Feng, C.; Wang, Y. Dynamics of fermentation parameters and bacterial community in high-moisture alfalfa silage with or without lactic acid bacteria. Microorganisms. 2021, 9, 1225. doi:10.3390/microorganisms9061225.
Point 2: Line 111 - How the color, smell and texture of SBM and FSBM were evaluated? please, specify.
Thank you for the reminder, we have added the corresponding information to the manuscript, please see Line 115-121. The specific methods are as follows:
The sensory evaluation was carried out with reference to the method of Meinlschmidt et al. [17]. After the FSBM samples were unsealed, sensory profiling was performed using descriptive sensory analysis: the samples were weighed into dishes, and the sensory evaluation team composed of 5 people to evaluate the colors, odor and hand feel characteristics of SBM and FSBM during fermented for 3 and 30 d, and aerobic exposure of 7 d by eye observation, nose smell and hand touch.
- Meinlschmidt, P.; Ueberham, E.; Lehmann, J.; Schweiggert-Weisz, U.; Eisner, P. Immunoreactivity, sensory and physico-chemical properties of fermented soy protein isolate. Food Chem. 2016, 205, 229-238. doi:10.1016/j.foodchem.2016.03.016.
Point 3: Line 211 - Please, add more information about the ANOVA analysis. Which parameters were significant? moreover, what was the R2, adjusted R2 and predicted R2, Adeq Precision and CV%?
Thank you for your good suggestions, and we have revised the data about the ANOVA analysis in manuscript, significance and R2, adjusted R2 and predicted R2, Adeq Precision and CV% all are shown in Table 1, the value of them were 0.8918, 0.827, 0.6249, 11.631, and 4.44, respectively.
Table 1. Analysis of variance for the quadratic polynomial mode
Source |
Sum of Squares |
df |
Mean Square |
F Value |
p-Value |
Model |
0.068 |
6 |
0.011 |
13.74 |
0.0003 |
A |
0.023 |
1 |
0.023 |
28.39 |
0.0003 |
B |
0.016 |
1 |
0.016 |
18.82 |
0.0015 |
C |
0.027 |
1 |
0.027 |
33.04 |
0.0002 |
AB |
1.53E-03 |
1 |
1.53E-03 |
1.85 |
0.2037 |
AC |
1.44E-03 |
1 |
1.44E-03 |
1.75 |
0.2159 |
BC |
3.04E-03 |
1 |
3.04E-03 |
3.68 |
0.0841 |
Residual |
8.27E-03 |
10 |
8.27E-04 |
|
|
Cor Total |
0.076 |
16 |
|
|
|
Std. Dev. |
0.029 |
R-Squared |
0.8918 |
|
|
Mean |
0.65 |
Adj R-Squared |
0.827 |
|
|
C.V. % |
4.44 |
Pred R-Squared |
0.6249 |
|
|
PRESS |
0.029 |
Adeq Precision |
11.631 |
  |
  |
Note: DF, degree of freedom; CV, coefficient of variation; A, glucose; B, peptone; C, inorganic salt.
